# Design and Application of Liquid Fertilizer pH Regulation Controller Based on BP-PID-Smith Predictive Compensation Algorithm

**Zihao Meng, Lixin Zhang \*, He Li, Runmeng Zhou, Haoran Bu, Yongchao Shan, Xiao Ma and Ruihao Ma**

School of Mechanical and Electrical Engineering, Shihezi University, Shihezi 832003, China; mzh9823@163.com (Z.M.); lihe@stu.edu.cn (H.L.); zhourunmeng@stu.edu.cn (R.Z.); buhaoran@stu.edu.cn (H.B.); shanyongchao@stu.edu.cn (Y.S.); maxiao@stu.edu.cn (X.M.); maruihao@stu.edu.cn (R.M.)
\* Correspondence: zhlx2001329@163.com

**Abstract:** The pH value of liquid fertilizer is a key factor affecting crop growth, so it is necessary to regulate its pH value. However, the pH regulation system has the characteristics of nonlinearity and time lag, which makes it difficult for the conventional controller to achieve accurate pH control. By analyzing the regulation process, this paper designs a BP-PID-Smith prediction compensator, which compensates for the error between the actual model and the theoretical model and improves the control accuracy. The pH regulation system with STM32F103ZET6 as the control core was also developed, and the performance tests were carried out under different flow rates to compare with the regulation system of PID-Smith and Smith algorithms. The experimental results showed that the maximum overshoot of the BP-PID-Smith prediction compensator was 0.27% on average, and the average adjustment time for pH value reduction from 7.5 to 6.8 was 71.39 s, which had good practicality and robustness to meet the actual control demand.

**Keywords:** water and fertilizer integration; pH adjustment; BP-PID-Smith algorithm; estimated compensation

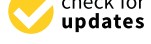



## 1. Introduction

In China, the use of traditional methods to irrigate crops is prone to the problem of water abuse and China's fertilizer use rate ranks first in the world all the year round, but the utilization rate of chemical fertilizers is not high. The integrated water and fertilizer technology is a new technology that combines irrigation and fertilization. According to the nutritional needs of crops, the pH value of liquid fertilizers is precisely regulated so that the roots of crops can fully absorb nutrients [1]. This technology can effectively reduce the pollution of fertilizers to soil and groundwater, and protect the ecological environment [2].

However, there is often a time lag in the process of pH adjustment, and the change in pH value of liquid fertilizer is also nonlinear. Therefore, how to quickly and accurately adjust the pH value of fertilizer to an appropriate range in the process of fertilization is the key research field of water fertilizer integration technology [3]. E. Ali et al. [4] proposed an adaptive PI algorithm that uses a simple process model to predict the pH closed-loop response and its sensitivity to PI parameter settings, and finally, the obtained information was directly used to adjust the PI controller parameters on-line. G.Balasubramanian et al. [5] proposed an adaptive control scheme based on Recurrent Neural Network (RNN). The scheme included an on-line adaptive RNN estimator and RNN controller, and the estimator weights were updated recursively by back-propagation algorithm. The controller weights were corrected by the steepest descent method. The proposed scheme was compared with the model-based IMC controller, and the results showed that the RNN-based controller had better performance in the nonlinear pH neutralization process.

Homero J. Sena et al. [6] used a real-time adaptive algorithm based on the Extended Kalman Filter (EKF) to correct the artificial neural network predictions at process runtime, which reduced the sum of squared errors in pH by 64.3% compared to the MPC of the artificial neural network without model adaptation. Douglas Alves Goulart and Renato Dutra Pereira [7] developed a Continuous Stirred-Tank Reactor (CSTR) neutralization simulator and an adaptive Particle Swarm Optimization (PSO) algorithm for automatic selection of Reinforcement Learning hyperparameters. During regulation and servo operation, the controller stabilized the effluent pH in the neutral range better than the PID controller. Shahin Salehi et al. [8] proposed an adaptive control scheme for pH value based on a fuzzy logic system and verified the effectiveness of the controller through simulation and experimental research. The results showed that the controller performed well in setpoint tracking and was much better than the PI controller. Hui Wu et al. [9] proposed a predictive control method based on Decentralized Fuzzy Inference (DFIPC), which locally linearized the nonlinear object model and predicted the future output of the control object according to its step response model. The method was applied to the pH neutralization process. The results showed that the method had better robustness than traditional model predictive control.

In the industrial field, Smith prediction compensation is mainly used to solve the pure lag of the system [10]. Guangda Chen [11] proposed a Smith predictor combined with Linear Active Disturbance Rejection Control (LADRC) and analyzed the stability of the Smith + LADRC time-delay control system from a theoretical point of view. Simulation and experimental results showed that the algorithm was superior to the traditional method in terms of overshoot and response time. Mahmoud Gamal et al. [12] combined the classical Smith predictor and the adaptive Smith predictor in a networked control system and compared it with other delay compensation schemes. The results showed that the scheme significantly improved the performance of the networked control system and reduced the impact of delay on the system. Chenkun Qi et al. [13] proposed a hybrid Smith predictor and phase lead compensation method. This method can achieve higher simulation fidelity with less convergence. The effectiveness of the compensation method was verified by the simulation of the undamped elastic contact process. Yonghui Nie et al. [14] proposed an optimal wide-area damping controller considering delay, using the Smith predictor to provide delay compensation and using particle swarm optimization to further improve the controller. The simulation results showed that the method improved the delay tolerance of the closed-loop system and improved the dynamic stability of the power system.

In this paper, according to the characteristics of pH value regulation of liquid fertilizers, the prediction compensation controllers based on Smith, PID-Smith and BP-PID-Smith were designed respectively. They were simulated and analyzed under the conditions of model matching and mismatching, and step response curves were obtained respectively. The performance of the three controllers was evaluated from four aspects: rise time, peak time, maximum overshoot, and adjustment time [15]. The results showed that the control effect of the BP-PID-Smith controller was the best. On this basis, an experimental platform was built to verify the practicability of the algorithm. The results showed that the BP-PID-Smith predictor compensator can effectively solve the adverse effects of the time delay and nonlinearity of the system in the fertilization process, and meet the control requirements for precise regulation of the pH value of liquid fertilizers.

The purpose of this paper is to design a BP-PID-Smith predictive compensation control algorithm, which can quickly adjust the pH value of water and fertilizer to the set value, and effectively solve the problems caused by factors such as time lag and nonlinearity in the pH adjustment process.

The contents of this paper are as follows: The first part introduces the research status of precise pH control and Smith prediction compensation. The second part explains the working principle of the pH regulation system, establishes the dynamic and static model of the pH regulation process, and reveals the nonlinear and time-delay characteristics of the pH regulation process. In the third part, the formulas of the Smith predictor compensator algorithm and BP neural network algorithm are derived, and the simulation models based

on Smith, PID-Smith, and BP-PID-Smith predictor compensation are established by using the Simulink module in MATLAB. In the fourth part, the above three models are simulated and analyzed, respectively, and the models are evaluated according to the results. In the fifth part, experiments are carried out to verify the practicability of the controller. The sixth part gives the conclusion.

## 2. Introduction of pH Value Regulation System and Analysis of Regulation Process

### 2.1. pH Control System Structure Composition

Figure 1 is the structural block diagram of the pH value regulation system of liquid fertilizer. The regulation system includes a water storage tank, solenoid valve, fertilizer tank, regulating liquid tank, flowmeter, pH value sensor, hose pump, mixing tank, and other main devices. The liquid in the water storage tank, fertilizer tank, and regulating liquid tank finally flows into the mixing tank and is stirred inside. A pH sensor is installed inside the mixing tank to monitor the pH value. The outlet of the mixing tank is connected with the field drip irrigation belt to transport the adjusted fertilizer to the field. The dilution ratio of liquid fertilizer is set to 1:8, the pH value of fertilizer before dilution is 7.5, and dilute hydrochloric acid with a concentration of 0.2 mol/L is used as the regulating liquid. Flowmeter and pressure gauge are installed at the inlet and outlet of the mixing tank. The hose pump is used as the conveying device of the regulating system. The three-phase asynchronous motor is connected with the pump body of the hose pump. The pressure formed by squeezing the hose by the roller is used to transport the materials. The system adjusts the outlet flow of the hose pump by changing the frequency of the frequency converter connected with the hose pump, to accurately adjust the pH value.

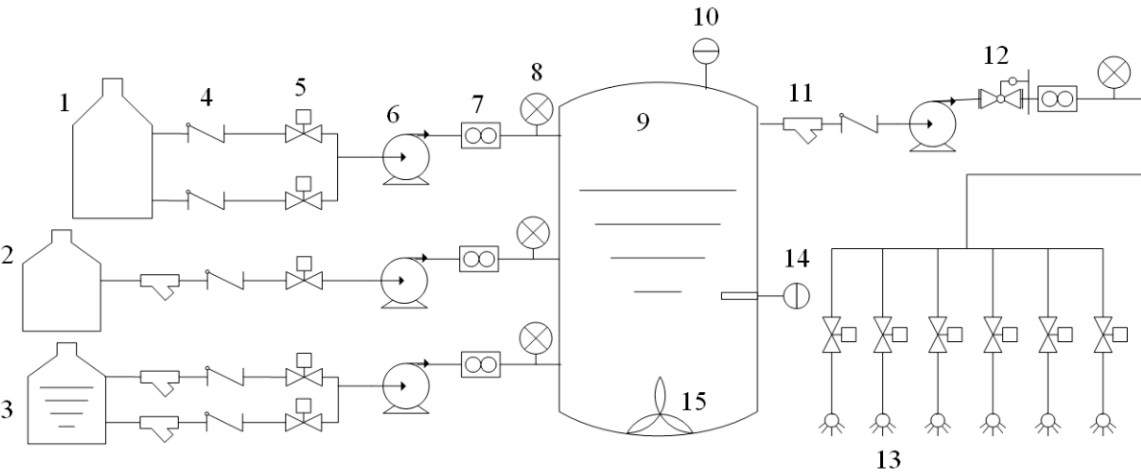

**Figure 1.** Structural block diagram of pH value regulation system of liquid fertilizer: 1, regulating liquid tank; 2, fertilizer tank; 3, water storage tank; 4, check valve; 5, solenoid valve; 6, hose pump; 7, flowmeter; 8, pressure gauge; 9, mixing tank; 10, liquid level gauge; 11, Y-type filter; 12, pressure holding valve; 13, drip irrigation belt; 14, pH sensor; 15, mixing pump.

In this paper, the STM32F103ZET6 microcontroller is used as the control core, and the BP-PID-Smith prediction compensation algorithm is written in it. The set pH value is used as the input value, and the pH value of the liquid fertilizer collected by the pH sensor is used as the actual feedback value for calculation. The required regulating liquid flow is calculated, and the flow is converted into the working frequency of the hose pump, to achieve the purpose of efficiently regulating the pH value of liquid fertilizer.

When the pH value regulation system of liquid fertilizer is working, the monitoring end will input the set pH value of liquid fertilizer and fertilization flow into the system, the hose pump at the fertilizer tank and the hose pump at the water storage tank will pump liquid fertilizer and water to the mixing tank in corresponding proportion for mixing. When the regulation system monitors the pH value in the mixing tank and the set value,

the solenoid valve opens and the hose pump at the regulating liquid tank starts to run to extract the regulating liquid into the mixing tank, and the stirring pump stirs the liquid in the mixing tank, and when the regulation system monitors the pH value of liquid fertilizer in the mixing tank reaches the set value, the system maintains a stable state.

### 2.2. Analysis of Liquid Fertilizer pH Regulation Process

Usually, water and liquid fertilizer are weakly basic, while the regulating liquid generally uses dilute hydrochloric acid. Therefore, the mixing of fertilizer, regulating liquid, and water can be considered a neutralization process of a strong acid and a weak base [16], which can be represented by a static pH equation describing the neutralization titration curve and a dynamic equation describing the state variables.

### 2.2.1. Objective Function and Design Variables

The ionization process of m-membered acid is described below.
First level ionization

$$H_m A \rightleftharpoons H_{m-1} A^- + H^+$$

$$K_{a1} = \frac{[H^+][H_{m-1}A^-]}{[H_m A]} \tag{1}$$

Secondary ionization

$$H_{m-1} A^- \rightleftharpoons H_{m-2} A^{2-} + H^+$$

$$K_{a2} = \frac{[H^+][H_{m-2}A^{2-}]}{[H_{m-1}A^-]} \tag{2}$$

$m$-level ionization

$$HA^{(m-1)-} \rightleftharpoons A^{m-} + H^+$$

$$K_{am} = \frac{[H^+][A^{m-}]}{[HA^{(m-1)-}]} \tag{3}$$

The ionization process of $n$-membered bases in liquid fertilizer can be expressed as:
First level ionization

$$B(OH)_n \rightleftharpoons B(OH)_{n-1}{}^+ + OH^-$$

$$K_{b1} = \frac{[B(OH)_{n-1}{}^+][OH^-]}{[B(OH)_n]} \tag{4}$$

Secondary ionization

$$B(OH)_{n-1}{}^+ \rightleftharpoons B(OH)_{n-2}{}^{2+} + OH^-$$

$$K_{b2} = \frac{[B(OH)_{n-2}{}^{2+}][OH^-]}{[B(OH)_{n-1}{}^+]} \tag{5}$$

$n$-level ionization

$$B(OH)^{(n-1)+} \rightleftharpoons B^{n+} + OH^-$$

$$K_{bn} = \frac{[B^{n+}][OH^-]}{[B(OH)^{(n-1)+}]} \tag{6}$$

where $K_{b1}, K_{b2}, \ldots, K_{bn}$ are the ionization equilibrium constants.

The ionization equilibrium of water can be expressed as:

$$H_2 O \rightleftharpoons H^+ + OH^-$$

$$K_w = [H^+][OH^-] \tag{7}$$

where $K_w = 10^{-14}$.

Let $x_i$ be the total ionic concentration of the acid or the total ionic concentration of the base in the fertilizer mixture, then

When $i$ is an acid:

$$x_i = [H_m A] + [H_{m-1} A^-] + \cdots + [A^{m-}] \tag{8}$$

When $i$ is a base:

$$x_i = [B(OH)_n] + [B(OH)_{n-1}{}^+] + \cdots + [B^{n+}] \tag{9}$$

Since the solution must always remain electrically neutral, the charge balance equation yields:

$$\sum_{i=acid} \left\{ [H_{p_i-1} A^-] + 2[H_{p_i-2} A^{2-}] + \cdots + p_i[A^{p_i-}] \right\} + [OH^-]$$
$$= \sum_{i=base} \left\{ \left[ B(OH)_{n_i-1}^+ \right] + 2\left[ B(OH)_{n_i-2}^{2+} \right] + \cdots + n_i[B^{n_i+}] \right\} + [H^+] \tag{10}$$

The pH equation can be derived from Equations (1)–(10) as:

$$\sum_{i=1}^{n} a_i([H^+]) x_i + [H^+] - \frac{K_w}{[H^+]} = 0 \tag{11}$$

When $i$ is an acid:

$$a_i([H^+]) = -\frac{m_i + (m_i - 1)\frac{[H^+]}{K_{am_i}} + \cdots + \frac{[H^+]^{m_i-1}}{K_{a2_i}K_{a3_i}\cdots K_{am_i}}}{1 + \frac{[H^+]}{K_{am_i}} + \cdots + \frac{[H^+]^{m_i-1}}{K_{a2_i}K_{a3_i}\cdots K_{am_i}} + \frac{[H^+]^{m_i}}{K_{a1_i}K_{a2_i}\cdots K_{am_i}}} \tag{12}$$

When $i$ is a base:

$$a_i([H^+]) = \frac{n_i[H^+]^{n_i} + (n_i - 1)\frac{K_w}{K_{bn_i}}[H^+]^{n_i-1} + \cdots + \frac{K_w^{n_i-1}[H^+]}{K_{b2_i}K_{b3_i}\cdots K_{bn_i}}}{[H^+]^{n_i} + \frac{K_w}{K_{bn_i}}[H^+]^{n_i-1} + \cdots + \frac{K_w^{n_i-1}[H^+]}{K_{b2_i}K_{b3_i}\cdots K_{bn_i}} + \frac{K_w^{n_i}}{K_{b1_i}K_{b2_i}\cdots K_{bn_i}}} \tag{13}$$

Since the whole pH neutralization process can be considered as the neutralization of a strong acid and a weak base, thus

$$a_1([H^+]) = -1 \tag{14}$$

$$a_2([H^+]) = \frac{[H^+]}{[H^+] + \frac{K_w}{K_b}} = \frac{1}{1 + \frac{[OH^-]}{K_b}} \tag{15}$$

The above analysis leads to the fact that the pH equation can be rewritten as:

$$-x_1 + \frac{1}{1 + 10^{pK_b + pH - 14}} x_2 + 10^{-pH} - 10^{pH-14} = 0 \tag{16}$$

where $pK_b = -\log K_b$, $x_1$, $x_2$ are the total ionic concentrations of acid and base in the fertilizer mix, respectively.

This equation is the static model of pH neutralization.

As can be determined from Equation (16), the control process of pH described by this static equation is strongly nonlinear.

### 2.2.2. Dynamic Model

Assuming that the liquid volume in the mixing tank is constant and uniformly mixed, ignoring the influence of liquid temperature on pH, the dynamic process of pH neutral-

ization reaction can be represented by a CSTR process [17–19]. The dynamic process of acid-base neutralization is shown in Figure 2.

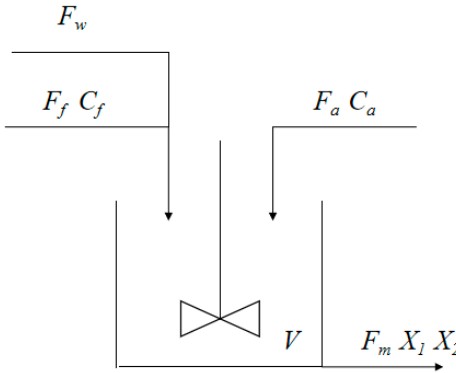

**Figure 2.** The dynamic process of acid-base neutralization.

In the figure, $V$ is the mixing tank volume; $F_w$ is the irrigation water flow rate; $C_w$ is the irrigation water input concentration; $F_a$ is the acid flow rate input into the mixing tank; $C_a$ is the regulating liquid input concentration; $F_f$ is the liquid fertilizer flow rate input into the mixing tank; $C_f$ is the liquid fertilizer input concentration; $F_m$ is the fertilizer mixture flow rate output from the mixing tank; $x_1$, $x_2$ are the acid and alkali concentrations in the output fertilizer mixture, respectively.

According to the principle of material conservation, the dynamic equation of each state variable is obtained when the fertilizer mixing process reaches equilibrium.

$$\begin{cases} V\frac{dx_1}{dt} = F_aC_a - F_mx_1 \\ V\frac{dx_2}{dt} = F_fC_f + F_wC_w - F_mx_2 \end{cases} \tag{17}$$

The input and output flows should be dynamically balanced, then:

$$F_m = F_a + F_w + F_f \tag{18}$$

Equations (16)–(18) together form a mathematical model of the pH neutralization process.

$$\begin{cases} -x_1 + \frac{1}{1+10^{pK_b+pH-14}}x_2 + 10^{-pH} - 10^{pH-14} = 0 \\ V\frac{dx_1}{dt} = F_aC_a - F_mx_1 \\ V\frac{dx_2}{dt} = F_fC_f + F_wC_w - F_mx_2 \\ F_m = F_a + F_w + F_f \end{cases} \tag{19}$$

From Equation (19), it can be seen that the mathematical model of the pH neutralization process consists of a static model and a dynamic model, and the dynamic process exhibits a slight nonlinearity, which is negligible when it is assumed that the input flow rate of the liquid fertilizer in the mixing tank is much larger than the input flow rate of the conditioning fluid.

However, Equation (16) expresses the inherent nonlinearity of the pH neutralization process. And in the actual fertilizer mixing process, besides the stirring and mixing process, the delay factors such as the slow flow of liquid in the pipeline and the time lag in the measurement link also have an impact on the pH regulation process.

Therefore, the pH regulation process is characterized by nonlinearity and time lag, which place higher demands on the performance of the controller.

2.2.3. Determination of the System Transfer Function

In this paper, the pH regulation system of liquid fertilizer is studied to analyze the process of pH regulation when mixing fertilizers.

The pH regulation characteristics and the complexity of the model are taken into account, and the mathematical model of the pH regulation system is described using a first-order system transfer function with a delay link [20].

$$G(s) = \frac{Ke^{-\tau s}}{Ts + 1} \tag{20}$$

A step response with a pH value of 6.8 was used as the input of the open-loop system. The sampling time interval of the system was set to 1 s. The initial pH value of the liquid in the mixing tank was 7.5, and the data of pH value changing with time was obtained. The first-order approximation method was used to input data into the computer and fit the step response curve of the system. The gain coefficient $K$ of the system is 1.02, the delay time is obtained $\tau$ is 7.5 and the time constant $T$ is 1.78. Therefore, the pH control process of liquid fertilizer has a time lag.

## 3. Design and Simulation of BP-PID-Smith Based Controller for pH Regulation System

### 3.1. Design of Time Lag Compensation for Smith's Prognosticator Model

PID control can adjust the size of the control quantity in time according to the error between the actual value and the desired value so that the actual value gradually approaches the desired value, which is a kind of closed-loop control with high reliability and robustness and is widely used in industry [21]. The PID closed-loop control system consists of two parts, the PID controller and the controlled object, as shown in Figure 3.

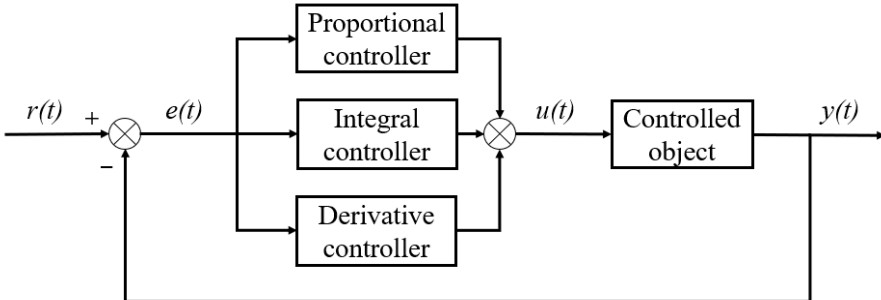

**Figure 3.** PID control structure diagram.

The error value $e(t)$ between the expected value $r(t)$ and the actual output value $y(t)$ is obtained; in the second step, the error values obtained are subjected to proportional, integral, and differential operations, and the closed-loop control quantity $u(t)$ is obtained after linear combination; in the third step, the controlled object receives the control quantity $u(t)$, and the output value $y(t)$ approaches the expected value $r(t)$ to complete the control of the controlled object within the error allowance. The mathematical expression of the control principle is:

$$u(t) = K_p \left[ e(t) + \frac{1}{T_i} \int_0^t e(\tau)d\tau + T_d \frac{de(t)}{dt} \right] \tag{21}$$

where $K_p$ is the proportionality constant, $T_i$ is the integration time constant, and $T_d$ is the differential time constant.

To discretize Equation (21), let $T$ be the sampling period, perform $k$ consecutive samples, and replace the continuous time $t$ with the discrete sampling time point $kT$

$$\begin{cases} t \approx kT \\ \int_0^t e(t)dt \approx T \sum_{j=0}^{k} e(jT) = T \sum_{j=0}^{k} e(j) \\ \frac{de(t)}{dt} \approx \frac{e(kT)-e[(k-1)T]}{T} \end{cases} \tag{22}$$

Bringing Equation (22) into Equation (21) and assuming that $T$ is sufficiently short, Equation (21) can be simplified to:

$$u(k) = K_p e(k) + K_i \sum_{j=0}^{k} e(j) + K_d[e(k) - e(k-1)] \tag{23}$$

where $K_p$, $K_i$, and $K_d$ are proportional, integral, and differential coefficients, $K_i = K_p \frac{T}{T_i}$, $K_d = K_p \frac{T_d}{T}$.

In the actual design of the controller, there are inevitably delays, including control delays and sensor delays, which may cause controller instability when the delays are relatively large. Therefore, the effect of delay needs to be considered when designing a controller. Smith predictor, as a classical solution, can offset and compensate for the delay effect of the system, significantly improve the control performance of the time-lag system, and reduce the instability of the system [22].

Smith's predictive compensation control structure is shown in Figure 4.

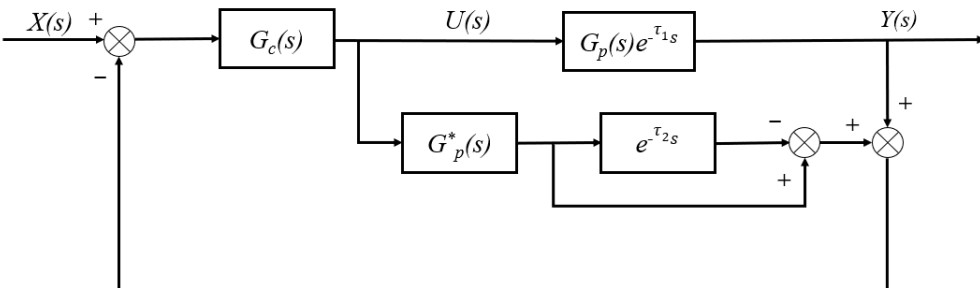

**Figure 4.** Smith prediction compensation control structure diagram.

Where $G_p^*(s)e^{-\tau_2 s}$ is the introduced Smith's prediction compensation transfer function, $G_p(s)e^{-\tau_1 s}$ is the system with delay model, and $G_c(s)$ is the main controller transfer function, when the model matches exactly, $G_p^*(s) = G_p(s)$ and $\tau_1 = \tau_2 = \tau$. The overall closed-loop transfer function of the system at this point is:

$$\frac{Y(s)}{X(s)} = \frac{G_c(s)G_p(s)e^{-\tau s}}{1 + G_c(s)G_p(s)} \tag{24}$$

The characteristic equation of the system is:

$$D(s) = 1 + G_c(s)G_p(s) = 0 \tag{25}$$

In this paper, the Ziegler–Nichols parameter rectification method is used for the initial rectification of the proportional, integral, and differential constants of the PID, as shown in Equation (26).

$$\begin{cases} K_P = 1.2\frac{T}{K \times \tau} \\ T_i = 2.2\tau \\ T_d = 0.5\tau \end{cases} \tag{26}$$

The controlled object model in this paper is shown in Equation (20), where $K = 1.02$, $T = 1.78$, $\tau = 7.5$, and the parameters are brought into Equation (26) to obtain the preliminary rectified values of the proportional, integral, and differential constants calculated by the Ziegler–Nichols method as $K_P = 0.28$, $K_i = 0.02$, $K_d = 1.05$, respectively.

The simulation model of the pH regulation system based on Smith's prediction compensation is shown in Figure 5.

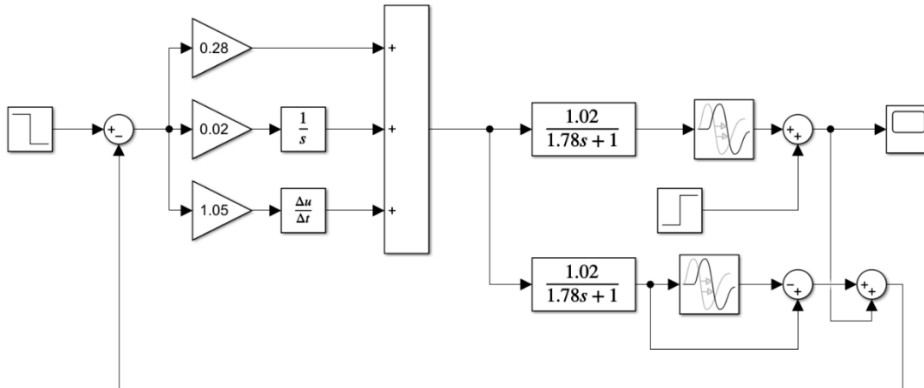

**Figure 5.** Simulation model of pH regulation system based on Smith's prediction compensation.

### 3.2. Design of PID-Smith Prediction Compensator

When the model cannot be matched exactly, at this time $G_p^*(s) \neq G_p(s)$, $\tau_1 \neq \tau_2$, the overall closed-loop transfer function of the system is:

$$\frac{Y(s)}{X(s)} = \frac{G_c(s)G_p(s)e^{-\tau_1 s}}{1 + G_c(s)\left[G_p^*(s) + G_p(s)e^{-\tau_1 s} - G_p^*(s)e^{-\tau_2 s}\right]} \tag{27}$$

From the above equation, it can be seen that when the deviation between the actual model and the theoretical model is large, there is a lag term in the characteristic equation of the system, which makes the Smith controller unable to correct the error between the actual model and the theoretical model even though, which may eventually cause the output signal of the system to oscillate and diverge. Therefore, Smith's prediction compensation is not suitable for the case where the theoretical model has a large deviation from the actual model.

The PID-Smith prediction control adds a suitable PID compensation controller to avoid the time lag term in the closed-loop characteristic equation and the model mismatch, which eventually leads to the oscillation and divergence of the output signal, thus reducing the effect of the time lag term on the system, and the structure of the PID-Smith prediction compensator is shown in Figure 6.

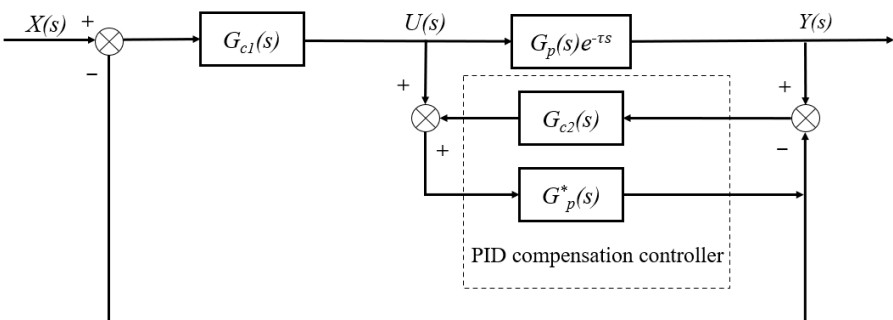

**Figure 6.** Structure of PID-Smith predictive compensator.

Where $G_{c2}(s)$ and $G_p^*(s)$ together form the compensation controller, $G_p(s)e^{-\tau s}$ is the system with delay model, and $G_{c1}(s)$ is the main controller model.

From Figure 6, the closed-loop transfer function of the system is:

$$\frac{Y(s)}{X(s)} = \frac{G_{c1}(s)G_p(s)e^{-\tau s}}{1 + G_{c1}(s)G_p^*(s)\frac{1+G_p(s)G_{c2}(s)e^{-\tau s}}{1+G_{c2}(s)e^{-\tau s}G_p^*(s)}} \tag{28}$$

The characteristic equation of the system is:

$$D(s) = 1 + G_{c1}(s)G_p^*(s)\frac{1 + G_p(s)G_{c2}(s)e^{-\tau s}}{1 + G_{c2}(s)e^{-\tau s}G_p^*(s)} = 0 \tag{29}$$

If the mode of $G_{c2}(s)$ is chosen to be small enough, then:

$$1 + G_p(s)G_{c2}(s)e^{-\tau s} \approx 1 \tag{30}$$

$$1 + G_{c2}(s)e^{-\tau s}G_p^*(s) \approx 1 \tag{31}$$

At this point the system characteristic equation simplifies to

$$D(s) = 1 + G_{c1}(s)G_p^*(s) = 0 \tag{32}$$

From the above equation, it can be seen that the system stability is not affected by the time lag of the compensating controller and the controlled object, and the system characteristic equation does not contain the time lag term.

The simulation model of the pH regulation system based on PID-Smith prediction compensation is shown in Figure 7.

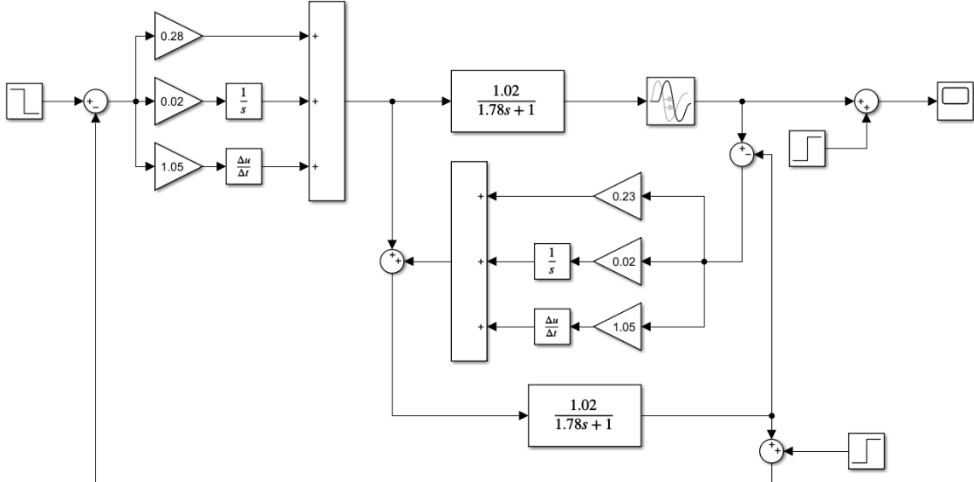

**Figure 7.** Simulation model of pH regulation system based on PID-Smith prediction compensation.

### 3.3. Design of BP-PID-Smith Prediction Compensator

Because the pH control system in this paper is a large time-delay system, and the traditional PID control is easily affected by time-delay, it cannot be optimized and adjusted according to the overall changes of the system. Therefore, this paper combines BP neural network with traditional PID control, designs a predictive compensator based on BP-PID-Smith, reduces the impact of time delay on the system, improves its learning efficiency by optimizing PID controller parameters, and realizes the parameter tuning of the control system.

The structure of the PID control system based on the BP neural network is shown in Figure 8.

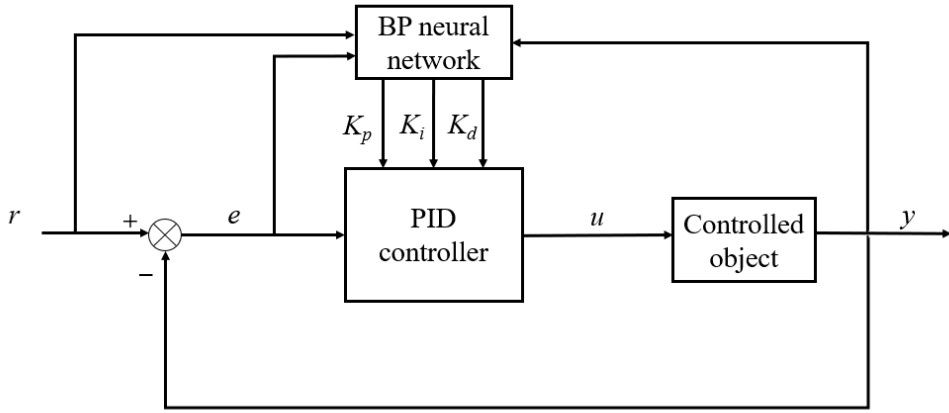

**Figure 8.** Structure of PID control system based on BP neural network.

The structure consists of a conventional PID and a BP neural network. The conventional PID realizes the closed-loop feedback control of the controlled object, and the BP neural network finally obtains the optimal PID control parameters of the system by continuously updating iterations according to the system state and learning algorithm. The neural network structure in Figure 8 is shown in Figure 9 below.

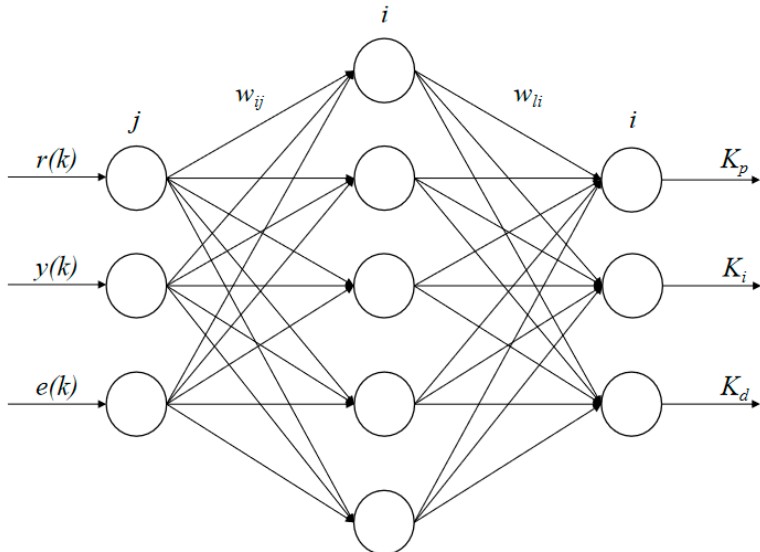

**Figure 9.** Optimization of PID parameters based on BP neural network.

The neural network consists of an input layer, an implicit layer, and an output layer, in which the input layer contains three neurons, and the inputs are $r(k)$, $y(k)$, and $e(k)$; to reduce the complexity of the system and improve the learning efficiency, the number of neurons in the implicit layer is set to five; the output layer contains three neurons, and the outputs correspond to the three parameters of the PID controller, $K_p$, $K_i$, and $K_d$, respectively.

A neural network structure based on 3-5-3, where the input layer inputs are:

$$O_j^{(1)} = x(j) \qquad (j = 1, 2, \ldots, M) \tag{33}$$

$M$ is the number of neurons in the input layer.

The implicit layer inputs and outputs are:

$$net_i^{(2)} = \sum_{j=0}^{M} w_{ij}^{(2)} O_j^{(1)} \tag{34}$$

$$O_i^{(2)}(k) = f\left(net_i^{(2)}(k)\right) \qquad (i = 1, 2, \ldots, Q) \tag{35}$$

where $Q$ is the number of neurons in the hidden layer and $w_{ij}^{(2)}$ is the hidden layer connection weight, $f(x) = \tanh(x) = \frac{e^x - e^{-x}}{e^x + e^{-x}}$.

The output layer inputs and outputs are:

$$net_l^{(3)}(k) = \sum_{i=0}^{Q} w_{li}^{(3)} O_i^{(2)}(k) \tag{36}$$

$$O_l^{(3)}(k) = g\left(net_l^{(3)}(k)\right) \quad (l = 1, 2, \ldots, N) \tag{37}$$

where $N$ is the number of neurons in the output layer and $w_{li}^{(3)}$ is the output layer connection weight, $g(x) = \frac{1}{2}[1 + \tanh(x)] = \frac{e^x}{e^x + e^{-x}}$.

The control quantity $u(t)$ of the PID controller is calculated according to Equation (23), $K_p$, $K_i$, and $K_d$ are $O_1^{(3)}(k)$, $O_2^{(3)}(k)$, and $O_3^{(3)}(k)$ as found in Equation (37). The selected performance metrics have the following functional form:

$$E(k) = \frac{1}{2}[r(k) - y(k)]^2 \tag{38}$$

The gradient descent method is used to continuously and iteratively update the connection weights between the neurons in the neural network so that the error signal decreases in the negative gradient direction. In addition, to speed up the convergence of the BP neural network algorithm and to obtain better dynamic properties, an inertia term is added to obtain a new update of the output layer connection weights when the learning rate is $\eta$.

$$\Delta w_{li}^{(3)}(k) = -\eta \frac{\partial E(k)}{\partial w_{li}^{(3)}} + \alpha \Delta w_{li}^{(3)}(k-1) \tag{39}$$

where $\alpha$ is the inertia factor.

$\frac{\partial E(k)}{\partial w_{li}^{(3)}}$ can be split into:

$$\frac{\partial E(k)}{\partial w_{li}^{(3)}} = \frac{\partial E(k)}{\partial y(k)} \times \frac{\partial y(k)}{\partial \Delta u(k)} \times \frac{\partial \Delta u(k)}{\partial O_l^{(3)}(k)} \times \frac{\partial O_l^{(3)}(k)}{\partial net_l^{(3)}(k)} \times \frac{\partial net_l^{(3)}(k)}{\partial w_{li}^{(3)}(k)} \tag{40}$$

After performing a series of simplifications, the updated output layer connection weights after learning by the neural network are obtained as:

$$\Delta w_{li}^{(3)}(k) = \alpha \Delta w_{li}^{(3)}(k-1) + \eta \delta_l^{(3)} O_i^{(2)}(k) \tag{41}$$

$$\delta_l^{(3)} = e(k) sgn\left(\frac{\partial y(k)}{\partial \Delta u(k)}\right) \frac{\partial \Delta u(k)}{\partial O_l^{(3)}(k)} g'\left(net_l^{(3)}(k)\right) \quad (l = 1, 2, \ldots, N) \tag{42}$$

Similarly, the update of the connection weights of the hidden layer after learning can be obtained as:

$$\Delta w_{ij}^{(2)}(k) = \alpha \Delta w_{ij}^{(2)}(k-1) + \eta \delta_i^{(2)} O_j^{(1)}(k) \tag{43}$$

$$\delta_i^{(2)} = f'\left(net_i^{(2)}(k)\right) \sum_{l=1}^{3} \delta_l^{(3)} w_{li}^{(3)}(k) \quad (i = 1, 2, \ldots Q) \tag{44}$$

The mathematical model for adding BP neural network is established above.

The PID main controller parameters are fine-tuned and the compensation controller is written using the BP-PID algorithm with the S-Function in the Simulink module, and the error between the actual model and the theoretical model is used as the input to the BP-PID algorithm.

The simulation model of the pH regulation system based on BP-PID-Smith prediction compensation is shown in Figure 10.

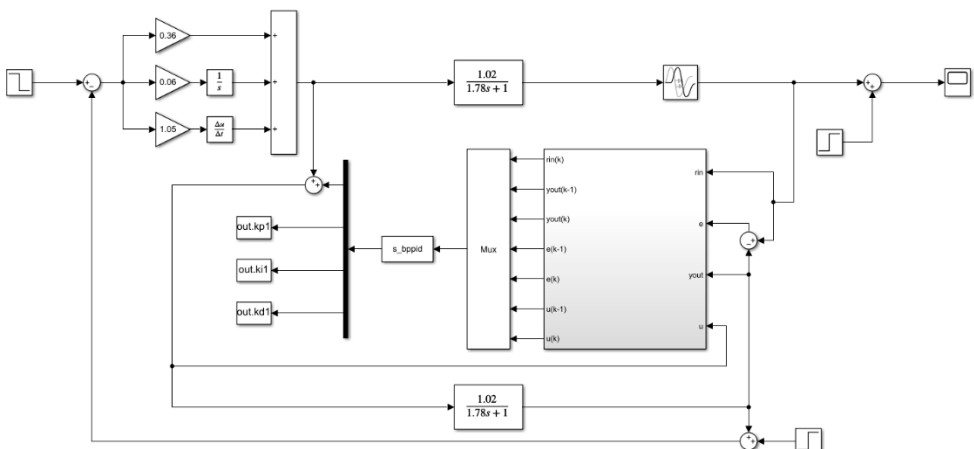

**Figure 10.** Simulation model of pH regulation system based on BP-PID-Smith prediction compensation.

## 4. Analysis of Simulation Results

### 4.1. Model Match

In this paper, based on BP neural network algorithm and Smith prediction compensation principle, the liquid fertilizer pH adjustment system based on the Smith prediction compensator, PID-Smith prediction compensator, and BP-PID-Smith prediction compensator were designed, respectively, and the final obtained simulation results were compared and analyzed. According to the actual situation, the initial pH value of liquid fertilizer was 7.5, the pH value of adjusted liquid fertilizer was set to 6.8, and the simulation time was set to 500 s. The focus of this paper is on the dynamic process of the system, and this process is described by the dynamic performance metrics, which are composed of rise time, peak time, regulation time, and overshoot of the controller. The rise time is the time required for the response to rise from zero to the final value for the first time; the peak time is the time required for the response to exceeding its final value to reach the first peak; the regulation time is the minimum time required for the response to reach and remain within $\pm 5\%$ (or $\pm 2\%$) of the final value; and the overshoot is the percentage of the ratio of the maximum deviation of the response $c(t_p)$ to the final value $c(\infty)$, i.e., $\sigma\% = \frac{C(t_p) - C(\infty)}{C(\infty)} \times 100\%$. The rise time, peak time and regulation time are important indicators to evaluate the response speed of the system, and the overshoot reflects the stability of the system control process. The response curves of the three controllers when the models are matched are shown in Figure 11, the training error curves of the BP-PID-Smith controller are shown in Figure 12, and the dynamic performance comparisons are shown in Table 1.

**Table 1.** Comparison of the dynamic performance of the three controllers for model match.

| Controller Type | Rise Time(s) | Peak Time(s) | Regulation Time (s) | Maximum Overshoot |
|---|---|---|---|---|
| Smith | 397.541 | 397.541 | 178.197 | 0 |
| PID-Smith | 115.902 | 163.947 | 83.770 | 0.13% |
| BP-PID-Smith | 56.712 | 66.891 | 36.230 | 0.03% |

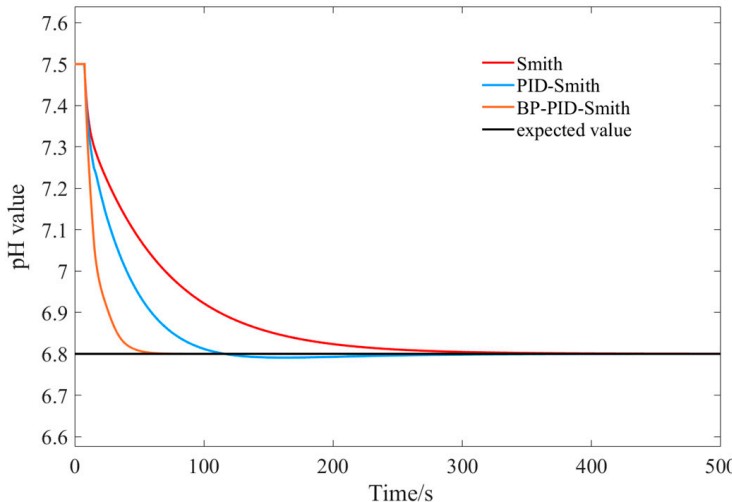

**Figure 11.** Step response curves of the three controllers under model matching.

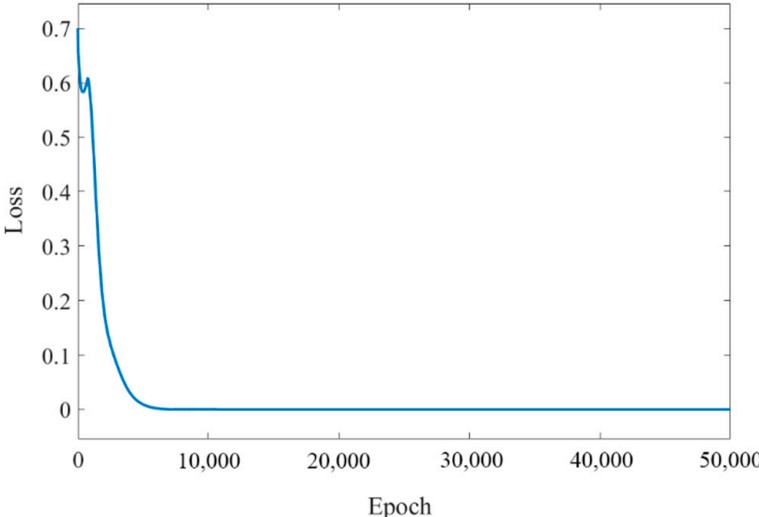

**Figure 12.** Training error curve of BP-PID-Smith controller under model matching.

According to the data in Table 1, the dynamic performance index of the BP-PID-Smith controller is significantly better than the other two controllers, so the BP-PID-Smith controller has a faster response and better steady-state performance compared with the PID-Smith controller and Smith controller. The Smith controller has a slow regulation time and rise time, reflecting its slow response time. The rise time and regulation time of the modified PID-Smith controller become shorter compared with the Smith controller, but at the same time, a certain amount of overshoot is generated, and the overall control effect is not as good as that of the BP-PID-Smith controller.

*4.2. Model Mismatch*

The above prediction compensator is based on the accurate mathematical model of the controlled object, however, in the actual process of pH regulation of liquid fertilizer, the accurate mathematical model of the system was difficult to obtain because of the complexity inside the system, which led to the actual model's inability to completely match with the theoretical controlled object model, thus seriously affecting the control effect of the controller and making the controller unable to operate stably. In this paper, three controllers were simulated under the conditions of model mismatch to examine the performance of the controllers.

In this paper, the actual model was set to $K = 1.5$, $T = 1.5$, $\tau = 12$, and a perturbation of amplitude 0.1 was added. The step response curves of the three controllers under the model mismatch condition are shown in Figure 13, the training error curves of the BP-PID-Smith controller are shown in Figure 14, and the dynamic performance comparison is shown in Table 2.

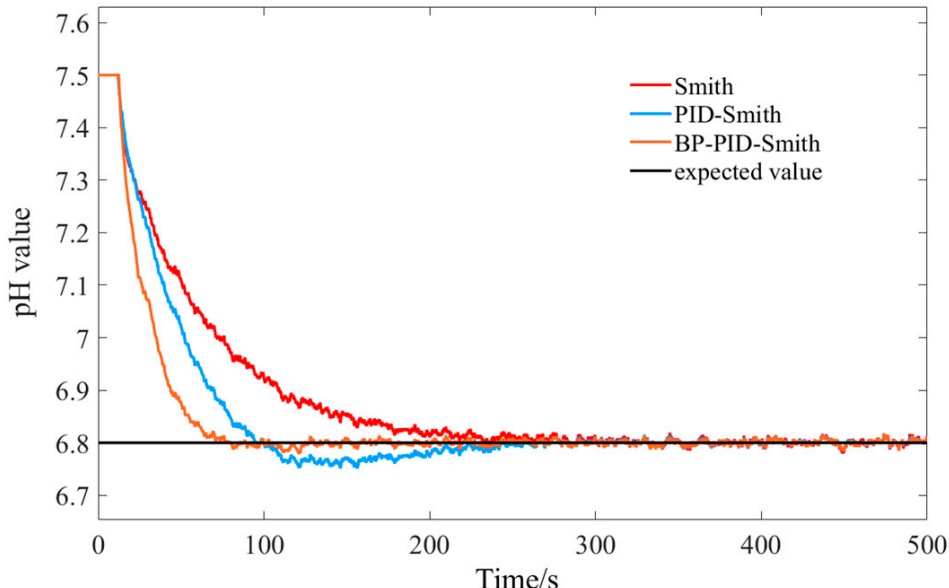

**Figure 13.** Step response curves of the three controllers under model mismatch.

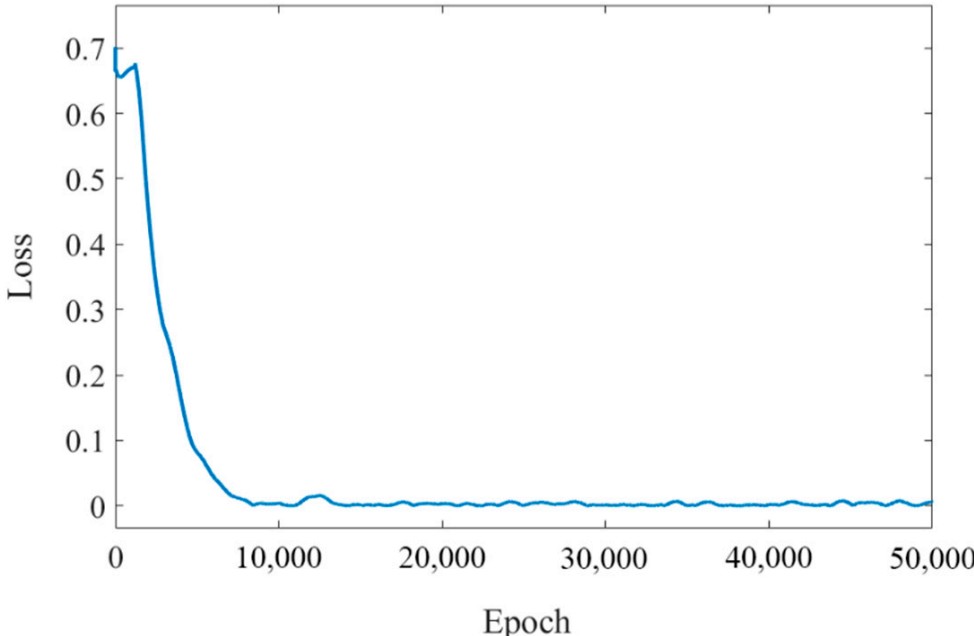

**Figure 14.** Training error curve of BP-PID-Smith controller under model mismatch.

**Table 2.** Comparison of the dynamic performance of the three controllers for the model mismatch.

| Controller Type | Rise Time(s) | Peak Time(s) | Regulation Time (s) | Maximum Overshoot |
|---|---|---|---|---|
| Smith | 232.228 | 449.377 | 170.216 | 0.28% |
| PID-Smith | 96.494 | 141.383 | 84.043 | 0.53% |
| BP-PID-Smith | 73.231 | 91.354 | 57.831 | 0.13% |

According to the data in Table 2, even though the Smith controller had a smaller maximum overshoot of 0.28%, its response speed was the slowest, the PID-Smith controller had a great improvement in response speed and stability compared to the Smith controller but there was still a higher overshoot, and the BP-PID-Smith controller had a better performance than the other two controllers both in terms of response speed and stability. The performance of the BP-PID-Smith controller was better than the other two controllers in terms of response speed and stability, and it had a smaller overshoot. Therefore, the BP-PID-Smith controller can meet the control requirements in the actual regulation process.

## 5. pH Regulation Experiment

### 5.1. Construction of the Experimental Platform for pH Regulation

In this paper, a pH regulation platform was built according to the system structure to further test the practicality of the BP-PID-Smith control algorithm. The STM32F103ZET6 microcontroller was used as the control core to receive the signal from the pH sensor using the I/O port, and the microcontroller adjusted the output frequency of the inverter by changing the magnitude of the analog voltage, thus changing the flow rate at the hose pump of the regulating liquid tank. The delivery flow rate of the hose pump was 1 m$^3$/h, rated power was 1.5 kW, and rated voltage was 380 V. The inverter's rated power was 2.2 kW, the output frequency can be adjusted from 0 to 400 Hz, rated voltage was 380 V. A pH sensor of RMD-ISSF-5 was used, with an accuracy of 0.01. The volume of the fertilizer mixture was kept at 40 L during the experiment. The pH adjustment experiment platform is shown in Figure 15.

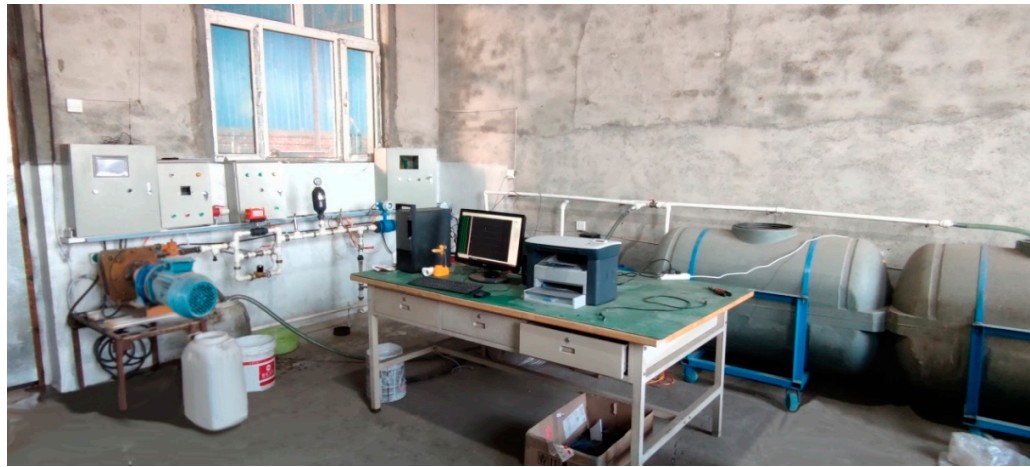

**Figure 15.** pH regulation experimental platform.

The regulation platform used USB-1252A to collect the data needed for the regulation process. The collector was equipped with an advanced measurement and control system with 16 analog input channels, 12-bit vertical resolution, and up to 500 kSa/s analog acquisition capability. The schematic diagram of the data acquisition and control system is shown in Figure 16.

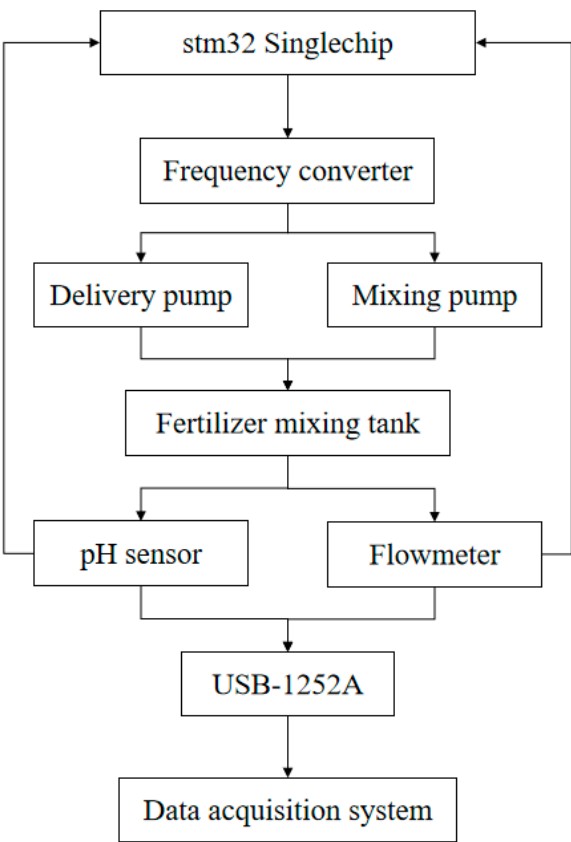

**Figure 16.** Schematic diagram of data acquisition and control system.

### 5.2. Analysis of Experimental Results

The fertilizer was diluted using dilute hydrochloric acid to maintain the pH of the fertilizer mixture at 7.5, and the flow rate of the hose pump at the fertilizer tank was adjusted to 0.35 m³/h, 0.58 m³/h, and 0.73 m³/h. The performance of the three controllers was tested. The experimental results are shown in Figures 17–19, and the performance indexes of the three controllers are shown in Tables 3–5.

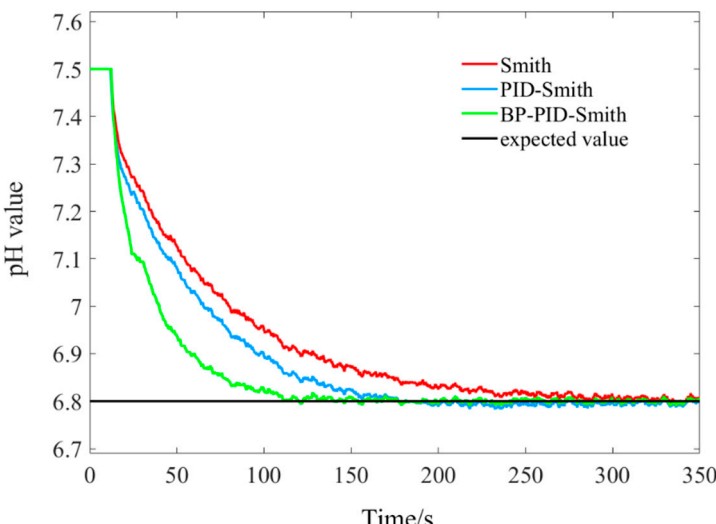

**Figure 17.** Regulation curves of three controllers for a fertilizer flow rate of 0.35 m³/h.

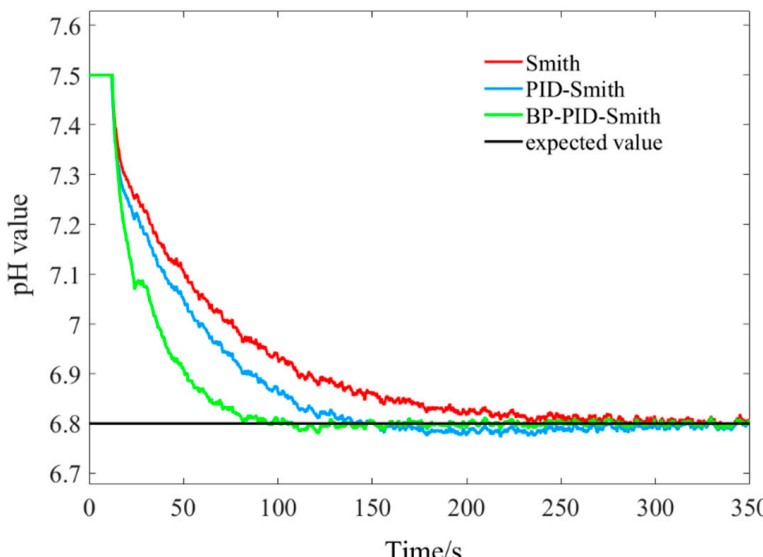

**Figure 18.** Regulation curves of three controllers for a fertilizer flow rate of 0.58 m$^3$/h.

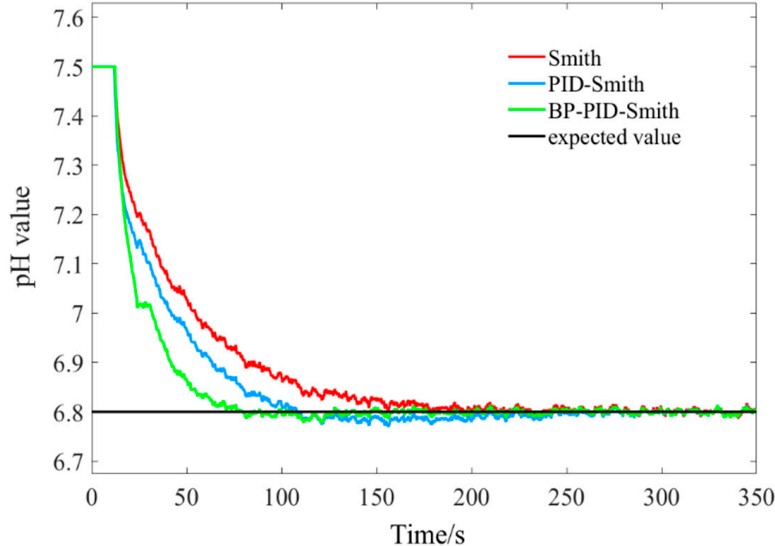

**Figure 19.** Regulation curves of the three controllers for a fertilizer flow rate of 0.73 m$^3$/h.

**Table 3.** Comparison of the dynamic performance of the three controllers at a fertilizer flow rate of 0.35 m$^3$/h.

| Controller Type | Rise Time (s) | Peak Time (s) | Regulation Time (s) | Maximum Overshoot |
|---|---|---|---|---|
| Smith | 285.395 | 333.145 | 178.644 | 0.06% |
| PID-Smith | 176.957 | 218.060 | 131.623 | 0.25% |
| BP-PID-Smith | 111.479 | 179.137 | 80.893 | 0.13% |

**Table 4.** Comparison of the dynamic performance of the three controllers at a fertilizer flow rate of 0.58 m$^3$/h.

| Controller Type | Rise Time (s) | Peak Time (s) | Regulation Time (s) | Maximum Overshoot |
|---|---|---|---|---|
| Smith | 235.547 | 288.110 | 174.819 | 0.05% |
| PID-Smith | 139.638 | 178.810 | 110.445 | 0.37% |
| BP-PID-Smith | 94.403 | 121.180 | 70.695 | 0.31% |

**Table 5.** Comparison of the dynamic performance of the three controllers at a fertilizer flow rate of 0.73 m$^3$/h.

| Controller Type | Rise Time(s) | Peak Time(s) | Regulation Time (s) | Maximum Overshoot |
|---|---|---|---|---|
| Smith | 178.436 | 235.560 | 111.320 | 0.02% |
| PID-Smith | 107.892 | 156.370 | 80.897 | 0.46% |
| BP-PID-Smith | 80.949 | 111.602 | 62.578 | 0.38% |

From Tables 3–5, it can be found that the performance of all three controllers improved as the flow rate of fertilizer increased. Although the Smith controller had the smallest overshoot, its rise time was slow and it could not respond quickly to the input pH setting value; the PID-Smith controller was a significant improvement in terms of rise time, peak time, and adjustment time compared with the Smith controller, but with the increase of flow rate, the overshoot amount also increased gradually. The BP-PID-Smith controller had the advantages of the other two controllers: it can reach the set pH value in a short time and the overshoot was also reduced compared with the PID-Smith controller; therefore, it can meet the control demand well even in the case of high flow rate.

## 6. Conclusions

For the liquid fertilizer pH regulation system, this paper fitted its mathematical model, got the transfer function of the pH regulation system, combined BP neural network algorithm with the Smith prognosticator, designed a BP-PID-Smith prognostication compensation controller, and compared the performance of three controllers, BP-PID-Smith, PID-Smith, and Smith, in both simulation and practical application. The results showed that the BP-PID-Smith predictive compensation controller was able to bring the pH to the set value at a faster rate in both cases with a smaller overshoot compared to the other two controllers.

According to the experiments, the BP-PID-Smith controller showed excellent dynamic performance at different fertilizer application flow rates and was able to respond to the input signal at a faster rate and achieve the desired target pH value. The average maximum overshoot was 0.27% and the average regulation time was 71.39 s, which was significantly better than the PID-Smith and Smith controllers.

The BP-PID-Smith predictive compensator can reduce the adverse effects of the system in the fertilization process due to time lag and nonlinearity in practical applications while possessing excellent dynamic performance and robustness to meet the control requirements in practical applications.

**Author Contributions:** This study was conceptualized by Z.M. and L.Z. The software was designed by Z.M. and validated by H.L., R.Z. and R.M. Z.M. provided resources and Z.M. curated data. The original draft of the manuscript was prepared by Z.M., Y.S. and X.M. reviewed and edited the manuscript. H.B., R.Z. and R.M. assisted with project administration. X.M. and H.L. managed funding acquisition. All authors have read and agreed to the published version of the manuscript.

**Funding:** This research was funded by the National Natural Science Foundation of China, grant number 52065055.

**Institutional Review Board Statement:** Not applicable.

**Informed Consent Statement:** Not applicable.

**Data Availability Statement:** All relevant data presented in the article are stored according to institutional requirements and, as such, are not available on-line. However, all data used in this Manuscript can be made available upon request to the authors.

**Conflicts of Interest:** The authors declare no conflict of interest.

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
