# Peer review of "Design and Application of Liquid Fertilizer pH Regulation Controller Based on BP-PID-Smith Predictive Compensation Algorithm"

_applsci, doi:10.3390/app12126162_

Round 1
Reviewer 1 Report
The authors paying attention to the problem that for exemple in China, the use of traditional methods to irrigate crops is prone to the problem of 28 water abuse, and China's fertilizer use rate ranks first in the world all the year round, but the utilization rate of chemical fertilizers is not high. They proposed an interesting solution. By analyzing the pH value regulation process, they designs a BP-PID-Smith prediction compensation algorithm, which combines the Smith prediction compensation algorithm with the BP-PID algorithm to compensate the error between the actual model and the theoretical model, so as to improve the control accuracy. When carrying out the analysis, the focus was on the achievements to date in this field, pointing to the pros and cons of the existing models. So that the work seems to have a solid exit base. The proposed model has been experimentally tested. The experimental results show that the average maximum overshoot of the BP- PID-Smith prediction compensation algorithm is 0.27%, and the average adjustment time for the pH value to drop from 7.5 to 6.8 is 71.39s, which is a satisfactory result in terms of control practice. The work includes the analysis of the dynamic and static model of the PH adjustment process, the derivation of the algorithm using neural networks and appropriate simulations and tests of the proposed models. Generally all variations are correctly describedI would like to draw your attention to the description of the K-factor in line 249. It is not so clear to the reader.
Reviewer 2 Report
Overall the entire paper needs to be reviewed by the authors in terms of English language mistakes including: use of hyphens, spelling and grammar especially incorrect use of tense.
Stronger statement needs to be made to highlight the novelty and relevance of this work. I am not convinced that a stand PID controller could not offer better performance if trained correctly.
Abstract Line 10-11, poor use of English and repetition.
Lines 13-16 poor phrasing and repetition.
The abstract should be re written with better English writing structure.
Line 36: often "a" time lag.
Should use hyphens in words when needed - Line 37.
39 was E.Ali the sole author?
Line 44 and elsewhere, replace “online” with “on-line”
Line 109, English needs improvement in the title.
Figure 1: Diagram messy, remove label lines and move label closer to components.
Line 153: “is described below”, remove the colon on line 154.
Line 204, space before new title.
Line 205 -208, sentence needs a rewrite.
Lines 217 - 220 - replicated
Line 224: Typo
All section titles should have a space before them to make them clearer to read.
Figure 6 - describe better in the text and diagram how Gp combines with X(s), no comparator shown.
Figure 8 - redraw to allocate more space for labels.
Line 340 - 345: I disagree that PID control would be “too simple” to successfully control the pH control system, perhaps the overall development of the control system in this paper over complicates the control requirements?
Table 1: missing a capital letter
What development software was used for the model development?
What was used for the NN development?
Line 473: notation problems.
Training development of the NN should be shown, with training graphs.
Would the reduced rise time of the proposed control systems actually offer real-world advantages? - How would the approx 100 s rise-time reduction benefit real-work applications?
The smith controller has significantly less overshoot, could that be indicative of a smaller Gain term which would also reduce the rise time for comparison?
Round 2
Reviewer 2 Report
Thank you for the updates
I would Like the development software stated in the text, ie Matlab, simulink module and the type of NN function used, as described in initial review.
You should show the training error graph for the ANN development, as described in initial review.
Other comments were handled by Authors.
